# Bayesian Active Meta-Learning under Prior Misspecification

Sabina J. Sloman [1]   Ayush Bharti [2]   Samuel Kaski [1] [2]

## Abstract

We study a setting in which an active meta-learner aims to separate the idiosyncracies of a particular task environment from information that will transfer between task environments. In a Bayesian setting, this is accomplished by leveraging a prior distribution on the amount of *transferable* and *task-specific information* an observation will yield, inducing a large dependency on this prior when data is scarce or environments change frequently. However, a misspecified prior can lead to bias in the inferences made on the basis of the resulting posterior — i.e., to the acquisition of *non-transferable information*. For an active meta-learner, this poses a dilemma: should they seek transferable information on the basis of their possibly misspecified prior beliefs, or task-specific information that enables better identification of the current task environment? Using the framework of Bayesian experimental design, we develop a novel diagnostic to detect the risk of non-transferable information acquisition, and leverage this diagnostic to propose an intuitive yet principled way to navigate the meta-learning dilemma — namely, seek task-specific information when there is risk of non-transferable information acquisition, and transferable information otherwise. We provide a proof-of-concept of our approach in the context of an experiment with synthetic participants.

## 1. Introduction

Most theoretical results in machine learning assume learning algorithms have access to substantial amounts of data from the environments in which they will be deployed. Yet the daunting task in the settings faced by many algorithms in practice is to bootstrap very little data from several different task environments to make robust predictions in new environments. This is the problem of transfer or *meta-learning*[1], a paradigm in which the learner's goal is to perform well in multiple different, but related, task environments (Pan & Yang, 2009; Kveton et al., 2021; Simchowitz et al., 2021). In the specific setting we study, each task environment the learner will encounter is characterized by both transferable effects (i.e., parameter values that are constant across task environments) and task-specific effects (i.e., parameter values that vary between task environments). The goal of the meta-learner is then to disentangle the transferable from the task-specific effects.

Our work is especially motivated by the paradigm of learning with human feedback: algorithms that rely on expert feedback, or "human-in-the-loop learning," are inherently constrained by limitations in the time, attention and knowledge of a given expert. While bringing multiple humans into the loop can overcome these limitations, eliciting feedback from more than one expert requires that the learner disentangle transferable knowledge shared between experts from the biases and idiosyncrasies of each.

Bayesian learning algorithms, which use prior information to make robust inferences in data-scarce settings, are a natural paradigm for such problems (Simchowitz et al., 2021). We analyze the setting of Bayesian meta-learning under prior misspecification: a learner specifies a prior distribution across a parameter space that includes both transferable and task-specific parameters, and this prior can diverge from the true parameter distribution. We leverage the framework of *Bayesian experimental design* (Rainforth et al., 2023; Valentin et al., 2023) to quantify the amount of information gained about the value of a transferable parameter. To the best of our knowledge, we are the first to use the framework of Bayesian experimental design to analyze the general setting of Bayesian meta-learning under prior misspecification (although related work has studied it for special cases; see Section 4). As we show in Section 1, misspecified prior information about the task-specific characteristics of a given task environment can affect the degree of **transferable information** (information that will transfer to new environments)

[1]Department of Computer Science, University of Manchester, Manchester, United Kingdom [2]Department of Computer Science, Aalto University, Helsinki, Finland. Correspondence to: Sabina J. Sloman <sabina.sloman@manchester.ac.uk>.

*Interactive Learning with Implicit Human Feedback Workshop at ICML 2023*, Honolulu, Hawaii, USA. PMLR 202, 2023. Copyright 2023 by the author(s).

[1]We use the convention that definitions in italics refer to terms that exist in the literature, while definitions in bold refer to novel terms introduced by this work.

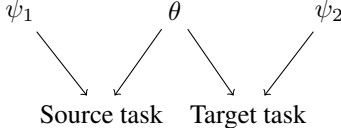

Source task   Target task

*Figure 1.* Problem setup. The value $\theta$ of the transferable parameter is shared across tasks, while the value of the task-specific parameter $\psi$ differs between tasks. The goal of the active meta-learner is to learn to take effective actions in the target task by actively eliciting data from the source task in order to maximize the information gained about the value of $\theta$.

a learner can acquire. In some cases, misspecified priors can even lead to **non-transferable information acquisition**, or to evidence in favor of the wrong transferable parameter value.

**Contributions.** In this paper, we leverage a Bayesian experimental design framework to analyze the setting of Bayesian meta-learning under prior misspecification. We introduce the meta-learning dilemma: should an active meta-learner seek transferable information on the basis of their possibly misspecified prior beliefs, or task-specific information that enables better identification of the current task environment? We introduce a credible lower bound on a measure of expected transferable information gain ($\text{CLB}_{\text{ETIG}}$), and demonstrate its use as a diagnostic of the risk of non-transferable information acquisition. We then demonstrate an application of $\text{CLB}_{\text{ETIG}}$ in the design of a Bayesian active learning algorithm for navigating the meta-learning dilemma.

**Setting.** Our setup is summarized in Figure 1. The meta-learner can take actions $x \in X$. Each action results in an outcome that can take values $y \in Y$, which follows a distribution determined by $x$ and the value $\omega \in \Omega$ of a set of unobservable parameters. $\Omega$ is partitioned into a set of **transferable parameters** $\theta \in \Theta$ and a set of **task-specific parameters** $\psi \in \Psi$. The values of the transferable parameters are stable across task environments, while the values of the task-specific parameters differ in each environment the learner will encounter. The learner's goal is to identify the true value of $\theta$, which we denote $\theta^*$. In each task that the learner encounters, a value of $\psi$ is drawn independently at random from a true distribution that characterizes the population of tasks. Together, a value $(x, \theta, \psi)$ induces a distribution over outcomes $p(y \mid x, \theta, \psi)$. This conditional distribution is assumed to be known to the learner.

However, the learner does not have access to the value $\theta^*$ or to the true distribution of $\psi$, and so assigns to it a prior distribution $p(\theta, \psi)$. We will hereafter use $q$ to refer to probability density functions induced by the true population distribution of $\psi$, which is unavailable to the learner. We

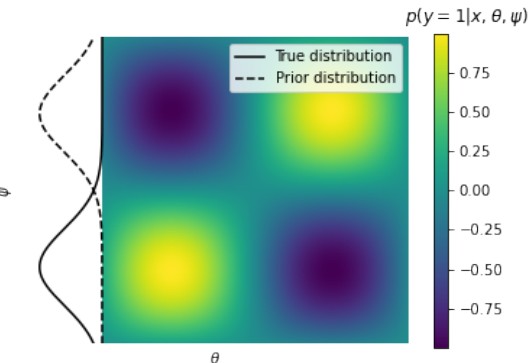

*Figure 2.* A misspecified prior can lead to non-transferable information acquisition. Under the learner's prior (dashed line), values of $\psi$ are concentrated in the upper half of the $y$-axis, and large values of $y$ occur most frequently when $\theta$ is large. Under the true distribution (solid line), values of $\psi$ are concentrated in the lower half of the $y$-axis. If the learner had access to this prior, they would consider observations of large values of $y$ as indicative of a small value of $\theta$.

reserve $p$ to refer to density functions used internally by the learning algorithm. The true distribution over outcomes induced by marginalizing over the parameter space can be written as $q(y \mid x) \coloneqq \int_{\Psi} p(y \mid x, \theta^*, \psi) \, q(\psi \mid \theta^*)$. The learner's expectations of the distribution over outcomes is $p(y \mid x) \coloneqq \int_{\Theta} \int_{\Psi} p(y \mid x, \theta, \psi) \, p(\psi \mid \theta) \, p(\theta)$.

After observing an action–outcome pair $(x, y)$, the learner updates the probability distribution $p(\theta, \psi)$ according to Bayes' rule, i.e., $p(\theta, \psi \mid y, x) = \frac{p(y \mid x, \theta, \psi) \, p(\theta, \psi)}{p(y \mid x)}$.

**Meta-learning dilemma.** To see how misspecification of the prior over $\psi$ can lead to non-transferable information acquisition, consider the toy example shown in Figure 2. The shading shows the probability $y = 1 \mid x$ under different values of $\theta$ and $\psi$. The dashed line shows a particular prior distribution one could place on $\psi$. Under this prior, if the learner observes $y = 1 \mid x$, this is considered evidence for a relatively high value of $\theta$. Conversely, if the learner observes $y = 0 \mid x$, this is considered evidence for a relatively low value of $\theta$. However, if the true distribution over $\psi$ is in fact characterized by the solid line, these observations would instead be indicative of a relatively low (high) value of $\theta$. In other words, the effect of misspecification over $\psi$ is to lead to initial convergence on a non-transferable value of $\theta$.

This example highlights a trade-off between learning task-specific and transferable information. Failing to invest resources in learning about $\psi$ would lead to wrong inferences about $\theta$. Yet if the learner invests too many resources in learning about $\psi$, they will not learn any information that will transfer to a new environment. This presents the learner

with a **meta-learning dilemma** which requires striking a balance between seeking transferable and task-specific information.

## 2. Sequential optimal experimental design

Sequential optimal experimental design (sOED) is an active learning paradigm in which the input $x$ is selected to maximize the mutual information between the observations it induces $y \mid x$, and the value of parameters $\omega$. sOED leverages the Bayesian modeling framework to define the mutual information measure, or *expected information gain* (EIG) associated with a possible input:

$$\text{EIG}(x) = \mathbb{E}_{\omega \sim p(\omega)} \left[ \mathbb{E}_{y \sim p(y \mid x, \omega)} \left[ \log \frac{p(\omega \mid y, x)}{p(\omega)} \right] \right]$$

$$= \mathbb{E}_{\omega \sim p(\omega)} \left[ \mathbb{E}_{y \sim p(y \mid x, \omega)} \left[ \underbrace{\log \frac{p(y \mid x, \omega)}{p(y \mid x)}}_{\text{Information gain}} \right] \right]$$

$$(1)$$

Notice that the EIG is necessarily a function of the learner's prior $p(y \mid x, \omega)$, since the true distribution $q(y \mid x, \omega)$ is unavailable in practice.

We refer to the inner term $\log \frac{p(\omega \mid y, x)}{p(\omega)} = \log \frac{p(y \mid x, \omega)}{p(y \mid x)}$ as the *information gain* about $\omega$: it is positive when $\omega$ is more likely under the posterior $p(\omega \mid y, x)$ than under the corresponding prior $p(\omega)$.

### 2.1. Expected Transferable and Task-specific Information Gain

The meta-learner's goal is to maximize information gain about the transferable parameters $\theta$. The more general setting of targeting inference at a subset of the parameters is referred to as the estimation of nuisance parameters (Foster et al., 2019) or of an embedded model (Rainforth et al., 2023).

We here distinguish this case by referring to the expected information gain measure computed with respect to $\theta$ as the **expected transferable information gain (ETIG)**:

$$\text{ETIG}(x) = \mathbb{E}_{\theta \sim p(\theta)} \left[ \mathbb{E}_{y \sim p(y \mid x, \theta)} \left[ \log \frac{p(y \mid x, \theta)}{p(y \mid x)} \right] \right]$$

$$= \mathbb{E}_{\theta, \psi \sim p(\theta, \psi)} \left[ \mathbb{E}_{y \sim p(y \mid x, \theta, \psi)} \left[ \underbrace{\log \frac{p(y \mid x, \theta)}{p(y \mid x)}}_{\text{Transferable Information Gain}} \right] \right]$$

$$(2)$$

where $p(y \mid x, \theta) = \mathbb{E}_{\psi \sim p(\psi \mid \theta)} [p(y \mid x, \theta, \psi)]$. This is not usually known in closed form, and so evaluating Equation (2) requires estimation of an implicit likelihood (Foster et al., 2019).

In the same way that we refer to the information gain of a particular set of values $(y \mid x, \omega)$, we also refer to the **transferable information gain (TIG)** of a set of values $(y \mid x, \theta)$. The TIG, or $\log \frac{p(\theta \mid y, x)}{p(\theta)} = \log \frac{p(y \mid x, \theta)}{p(y \mid x)}$, is the amount an observation $y \mid x$ increases the learner's beliefs in the value $\theta$. The interpretation parallels that of the information gain in the context of the EIG: TIG is positive when $\theta$ is more likely under the posterior $p(\theta \mid y, x)$ than under the corresponding prior $p(\theta)$.

We define the **task-specific information gain** and **expected task-specific information gain** analogously, i.e., as the information an observation provides about a value $\psi$ and as the learner's expectation of the task-specific information gain across their prior, respectively:

$$\text{ETSIG}(x) = \mathbb{E}_{\psi \sim p(\psi)} \left[ \mathbb{E}_{y \sim p(y \mid x, \psi)} \left[ \log \frac{p(y \mid x, \psi)}{p(y \mid x)} \right] \right]$$

$$= \mathbb{E}_{\theta, \psi \sim p(\theta, \psi)} \left[ \mathbb{E}_{y \sim p(y \mid x, \theta, \psi)} \left[ \underbrace{\log \frac{p(y \mid x, \psi)}{p(y \mid x)}}_{\text{Task-specific Information Gain}} \right] \right]$$

$$(3)$$

## 3. Sequential optimal experimental design for meta-learning

We now leverage the sOED framework to make more precise the nature of the meta-learning dilemma and develop tools to navigate it. Section 3.1 introduces the concept of actual information gain, which will allow us to formalize the concepts of transferable vs. non-transferable information acquisition. Section 3.2 introduces a novel diagnostic for detecting the risk of non-transferable information acquisition. Section 3.3 introduces an application of this diagnostic to the construction of an sOED method to trade-off between seeking transferable and task-specific information in a principled way. Section 3.4 gives details of a method to estimate the ETIG and ETSIG. Section 5 will then combine these components to provide a proof-of-concept implementation of our proposed approach to navigating the meta-learning dilemma in the context of an experiment designed to learn transferable information from the behavior of several synthetic participants.

### 3.1. Transferable vs. non-transferable information

EIG and ETIG give the expectation of the relevant information gain measure under the learner's prior. In the case of prior misspecification, these may diverge from the *actual* average information gain the learner would achieve in multiple task environments drawn from the true distribution.

This motivates our measure of **actual information gain (AIG)**:

$$\text{AIG}(x) = \mathbb{E}_{\omega \sim q(\omega)} \left[ \mathbb{E}_{y \sim q(y \mid x, \omega)} \left[ \log \frac{p(y \mid x, \omega)}{p(y \mid x)} \right] \right] \tag{4}$$

which differs from the EIG (Equation (1)) in that the outer distribution $q(\omega, y \mid x)$ is the true distribution.

Equation (4) can be rewritten as

$$\text{AIG}(x) = \mathbb{E}_{\omega \sim q(\omega)} \left[ \text{D}_{\text{KL}} \left( q(y \mid x, \omega) \,\|\, p(y \mid x) \right) - \text{D}_{\text{KL}} \left( q(y \mid x, \omega) \,\|\, p(y \mid x, \omega) \right) \right] \tag{5}$$

The derivation of Equation (5) is given in Appendix A. This can yield insights into when the learner should expect gains in transferable information even under prior misspecification: in the case where the likelihood is known, the term $\text{D}_{\text{KL}} \left( q(y \mid x, \omega) \,\|\, p(y \mid x, \omega) \right)$ is zero, and the AIG reduces to an expectation over a Kullback-Leibler divergence and thus must be non-negative — indicating acquisition of transferable information.

Like the EIG, the AIG can be straightforwardly extended to the meta-learning case. The **Actual Transferable Information Gain (ATIG)** is

$$\text{ATIG}(x) = \mathbb{E}_{\theta \sim q(\theta)} \left[ \mathbb{E}_{y \sim q(y \mid x, \theta)} \left[ \log \frac{p(y \mid x, \theta)}{p(y \mid x)} \right] \right]$$
$$= \mathbb{E}_{\theta \sim q(\theta)} \left[ \text{D}_{\text{KL}} \left( q(y \mid x, \theta) \,\|\, p(y \mid x) \right) - \text{D}_{\text{KL}} \left( q(y \mid x, \theta) \,\|\, p(y \mid x, \theta) \right) \right]. \tag{6}$$

Here, the likelihood $p(y \mid x, \theta)$ can also be misspecified: $\mathbb{E}_{\psi \sim q(\psi \mid \theta)} \left[ q(y \mid x, \theta, \psi) \right] = \mathbb{E}_{\psi \sim q(\psi \mid \theta)} \left[ p(y \mid x, \theta, \psi) \right]^2$ is not necessarily equal to $\mathbb{E}_{\psi \sim p(\psi \mid \theta)} \left[ p(y \mid x, \theta, \psi) \right]$. Thus, unlike in the standard case, the second term inside the expectation (here, $\text{D}_{\text{KL}} \left( q(y \mid x, \theta) \,\|\, p(y \mid x, \theta) \right)$) does not drop out, yielding no general guarantees about the sign of the ATIG. In other words, a learner operating under a misspecified prior may acquire evidence for the wrong value of the transferable parameter. We refer to this as acquiring **non-transferable** (as opposed to **transferable**) **information**. The sign of the ATIG can be interpreted as an indicator of whether the learner is acquiring transferable (positive ATIG) or non-transferable (negative ATIG) information.

Equation (6) also reveals the nature of the dilemma the meta-learner faces: not only are they tasked with identifying designs that lead each transferable parameter value to make distinct predictions (maximizing $\text{D}_{\text{KL}} \left( q(y \mid x, \theta) \,\|\, p(y \mid x) \right)$), but simultaneously with learning what those predictions are in the first place (minimizing $\text{D}_{\text{KL}} \left( q(y \mid x, \theta) \,\|\, p(y \mid x, \theta) \right)$).

---

[2]The equality follows since the distribution $q(y \mid x, \omega)$ is, by assumption, available to the learner.

## 3.2. ETIG credible lower bound

Of course, the learner does not have access to the true distribution and so cannot compute the ATIG. Here, we introduce the **ETIG credible lower bound** ($\text{CLB}_{\text{ETIG}}$), a diagnostic that can be computed without knowledge of the true distribution but which can nevertheless indicate when the learner may be in danger of acquiring non-transferable information.

Notice that the ETIG (Equation (2)) and ATIG (Equation (6)) are both written as an expectation over the TIG, where the ETIG is evaluated by computing the expectation with respect to the learner's prior $p$, and the ATIG is evaluated by computing the expectation with respect to the true distribution $q$. In other words, both can be estimated by computing the TIG corresponding to a sufficiently large number of combinations $(\theta, y \mid x)$ and then taking an average weighted by the appropriate distribution. Given infinite samples of $(\theta, y \mid x)$ and the corresponding TIG values, one of these samples would correspond to the actual (as yet undetermined) future observation $y \mid x$ and true parameter value $\theta^*$. The corresponding TIG value, which we refer to as TIG*, would indicate the amount of transferable information gain the learner would obtain after collecting their data. Quantifying the risk of non-transferable information acquisition then amounts to determining how likely it is that TIG* is negative.

The intuition behind our approach is to leverage the empirical distribution of TIG values that the learner computes to estimate the ETIG in order to additionally compute a credible lower bound on the value of TIG*. The ETIG is estimated as the mean of this empirical distribution of TIG values, and the **ETIG credible lower bound** ($\text{CLB}_{\text{ETIG}}$) can be interpreted as the lower bound of a credible interval constructed around that mean. If the value of $\text{CLB}_{\text{ETIG}}$ at a particular value of $x$ is negative, that indicates that collecting observations induced by $x$ poses a risk of non-transferable information acquisition.

More specifically, the $\text{CLB}_{\text{ETIG}}$ is computed as:

$$\text{CLB}_{\text{ETIG}}(x) = \mathbb{E}_{\theta, y \mid x \sim p(\theta, y \mid x)} \left[ \log \frac{p(y \mid x, \theta)}{p(y \mid x)} \right]$$
$$- \beta \sqrt{\text{Var}_{\theta, y \mid x \sim p(\theta, y \mid x)} \left[ \log \frac{p(y \mid x, \theta)}{p(y \mid x)} \right]} \tag{7}$$

The margin $\beta$ determines the size of the credible interval: larger margins will lead to lower $\text{CLB}_{\text{ETIG}}$ values, i.e., a lower implied risk tolerance for non-transferable information acquisition. Section 3.3 discusses methods for selecting $\beta$ in the context of our proposed algorithm.

We here briefly discuss how $\text{CLB}_{\text{ETIG}}$ connects to existing measures and frameworks for similar problems.

**Interpretation as an exploration penalty.** Related to the meta-learning dilemma is the exploration–exploitation dilemma, where online learners face a trade-off between exploiting (possibly misspecified) prior information about high-valued actions and exploring to gather observations that will refine this prior information. In the context of sequential optimization algorithms (e.g., Bayesian optimization; see Section 4), a common way to navigate the exploration–exploitation dilemma is to define an upper-confidence bound (UCB) measure which is an additive combination of the mean of the learner's distributional prediction at a given input and the standard deviation around that mean (Schulz et al., 2018). A UCB measure can be interpreted as an optimistic, or plausible upper bound on the, prediction at a given point.

$\text{CLB}_{\text{ETIG}}$ is computed similarly to a UCB measure, with the critical difference that a multiplier of the standard deviation is *subtracted* from, rather than added to, the mean prediction: unlike UCB, $\text{CLB}_{\text{ETIG}}$ is risk-*averse* in the sense that an input that induces a high standard deviation in TIG values is penalized rather than favored. In our framework, the standard deviation in TIG values is interpreted as the degree of risk of a non-transferable information gain (a negative characteristic of an input) rather than of uncertainty that can be reduced (a positive characteristic in an exploration framework). While UCB measures can be interpreted as optimistic estimates of a learner's predictions, $\text{CLB}_{\text{ETIG}}$ can instead be thought of as a pessimistic estimate of the ETIG.

**Interpretation as a robustness measure.** *Robust expected information gain (REIG)* is the EIG of the worst-case distribution inside an ambiguity set characterized by a pre-specified radius around the prior distribution (Go & Isaac, 2022). Under certain conditions, $\text{CLB}_{\text{ETIG}}$ can be interpreted similarly to the REIG, where the user specifies the parameter $\beta$ rather than the radius of the ambiguity set; see Appendix B for additional details. Intuition for this can be established by realizing that a worst-case measure can be thought of as a pessimistic estimate of the ETIG, and thus as an exploration penalty.

### 3.3. Proposed algorithm

We here describe an application of $\text{CLB}_{\text{ETIG}}$ as a component of an sOED algorithm to navigate the meta-learning dilemma, i.e., balance the acquisition of transferable and task-specific information. We refer to this approach as sOED for meta-learning (sOED-ML). sOED-ML can be summarized by a simple heuristic: seek task-specific information when there is risk of non-transferable information acquisition, and transferable information otherwise.

sOED-ML relies on the $\text{CLB}_{\text{ETIG}}$ to detect the risk of non-

---

**Algorithm 1** Sequential optimal experimental design for meta-learning (sOED-ML)

**Input:** Candidate designs $X$, prior distribution $p(\omega, y \mid X)$, margin $\beta$, number of Monte Carlo samples $N$

**Output:** Optimal design $x^* \in X$

Sample $\{(\theta_1, \psi_1, y_1) \dots (\theta_N, \psi_N, y_N)\} \sim p(\theta, \psi, y \mid X)$
Compute $\text{TIG}(\theta_i, y_i, X) \,\forall\, (\theta_i, y_i)$
$\text{ETIG}(X) \leftarrow \mathbb{E}_{\theta_i, y_i} [\text{TIG}(\theta_i, y_i, X)]$
$\text{SDTIG}(X) \leftarrow \sqrt{\text{Var}_{\theta_i, y_i} [\text{TIG}(\theta_i, y_i, X)]}$
$\text{CLB}_{\text{ETIG}}(X) \leftarrow \text{ETIG}(X) - \beta \text{SDTIG}(X)$
$i^* \leftarrow \text{argmax}_i \, \text{ETIG}(X)[i]$
$x^* \leftarrow X[i^*]$
**if** $\text{CLB}_{\text{ETIG}}(x^*) < 0$ **then**
    Compute $\text{TSIG}(\psi_i, y_i, X) \,\forall\, (\psi_i, y_i)$
    $\text{ETSIG}(X) \leftarrow \mathbb{E}_{\psi_i, y_i} [\text{TSIG}(\psi_i, y_i, X)]$
    $i^* \leftarrow \text{argmax}_i \, \text{ETSIG}(X)[i]$
    $x^* \leftarrow X[i^*]$
**end if**
**return** $x^*$

---

transferable information acquisition. More specifically, we consider there to be a risk of non-transferable information acquisition if $\text{CLB}_{\text{ETIG}}$ is negative at the ETIG maximizer — in other words, if pursuing a standard version of sOED would lead to a substantial risk of non-transferable information acquisition. If this condition is met, sOED-ML selects the input that maximizes the ETSIG. Otherwise, it selects the input that maximizes the ETIG. The algorithm is given in Algorithm 1.

The margin $\beta$ is considered a hyperparameter of sOED-ML. In Section 5, we empirically explore implementations in which $\beta$ decays as the experiment progresses. The implied higher tolerance for the risk of non-transferable information indicates both a lower risk of misspecification as additional data is collected, and the reduced benefit of seeking task-specific information as the data budget from a given task becomes exhausted.

### 3.4. Estimating ETIG and ETSIG

Estimating the EIG is non-trivial, as it requires computing either the posterior $p(\theta, \psi \mid y, x)$ or the marginal distribution $p(y \mid x)$, neither of which are in general known in closed form (Foster et al., 2019). Estimating the ETIG and ETSIG is even more complicated: as discussed in Section 2.1, these require an additional approximation to the likelihood $p(y \mid x, \theta)$ itself (Foster et al., 2019).

While the TIG and task-specific information gain can be computed as the log ratio of conditional and marginal likeli-

hoods ($\log \frac{p(y \mid x, \theta)}{p(y \mid x)}$ and $\log \frac{p(y \mid x, \psi)}{p(y \mid x)}$, respectively) rather than as the log ratio of posterior and prior parameter likelihoods, a representation of the posterior parameter distribution is nevertheless required in the context of sOED, where the posterior from one experimental trial is used as the prior on the following trial. Thus in our setting, the prior distribution across which the conditional and marginal likelihoods, and the expectations of the TIG and task-specific information gain themselves, is computed is not usually known in closed form.

One way to circumvent the intractability of the parameter distribution is to approximate it with the best-fitting member of a family of variational distributions (Foster et al., 2019). A common choice of variational distribution is a multivariate normal, which is theoretically justified as the asymptotic parameter distribution in the limit of infinite data (Paninski, 2005).

To estimate the ETIG and ETSIG in the context of the simulation experiments reported in Section 5, we approximate the prior distribution as a multivariate normal and then apply importance weights to correct for biases induced by the approximation[3]. Using a multivariate normal allows us to leverage analytical formulations for conditional parameter distributions when estimating the likelihood. We will use $\tilde{p}$ to refer to probability density functions induced by the multivariate approximation to the prior.

Of course, the quality of the ETIG and ETSIG estimates depends on the quality of the variational approximation. If the approximation is biased (in our case, if the true distribution is not a multivariate normal), this will lead to biased estimates of the ETIG and ETSIG. We follow previous work and sample from the variational prior, but correct for bias in this approximation using importance weights (Ryan et al., 2015; Foster et al., 2019; Senarathne et al., 2020).

Approximating the prior with a multivariate normal, as opposed to another variational family, offers two practical advantages:

1. Bias in the variational approximation can lead to underrepresentation of parameter values that fall in the tails of the true prior. The multivariate normal includes the variance of the distribution as an explicit parameter, and so can be straightforwardly rescaled to accommodate better coverage of the tails of the posterior. Following Ryan et al. (2015), to construct our variational approximation we inflate the covariance of the

best-fitting distribution, and then correct for bias using importance weighting.

2. The conditional variational prior is known in closed form, which allows fast sampling from variational approximations to $p(\psi \mid \theta)$ and $p(\theta \mid \psi)$.

Using this method, the estimates of the ETIG and ETSIG corresponding to a given input $x$ are computed as follows:

$$\widehat{\text{ETIG}}(x) =$$
$$\sum_{i=1}^{N} w(\theta_i, \psi_i) \log \frac{\sum_{j=1}^{M} w(\psi_j \mid \theta_i)\, p(y_i \mid x, \theta_i, \psi_j)}{\sum_{l=1}^{M} w(\theta_l, \psi_l)\, p(y_i \mid x, \theta_l, \psi_l)} \quad (8)$$

$$\widehat{\text{ETSIG}}(x) =$$
$$\sum_{i=1}^{N} w(\theta_i, \psi_i) \log \frac{\sum_{j=1}^{M} w(\theta_j \mid \psi_i)\, p(y_i \mid x, \theta_j, \psi_i)}{\sum_{l=1}^{M} w(\theta_l, \psi_l)\, p(y_i \mid x, \theta_l, \psi_l)}. \quad (9)$$

Samples subscripted by $i$ are drawn from the variational prior $p(y \mid x, \theta, \psi)\, \tilde{p}(\theta, \psi)$[4]. Samples subscripted by $l$ are drawn from the variational prior $\tilde{p}(\theta, \psi)$. Samples subscripted by $j$ are drawn from the conditional variational prior $\tilde{p}(\psi \mid \theta_i)$ (in Equation (8)) or $\tilde{p}(\theta \mid \psi_i)$ (in Equation (9)), which can be found in analytical form by conditioning the multivariate normal on $\theta_i$ (Equation (8)) or $\psi_i$ (Equation (9)).

$w$ indicates the relevant importance weighting function. For example, $w(\theta_i, \psi_i)$ is computed as:

$$w(\theta_i, \psi_i) = \frac{p(\theta_i, \psi_i)/\tilde{p}(\theta_i, \psi_i)}{Z^N}, \quad (10)$$

where $Z^N$ is a normalizing constant that ensures the $N$ importance weights sum to 1.

## 4. Related literature

Our work is closely related to work on the effect of misspecified priors on the performance of Bayesian decision-making algorithms (Simchowitz et al., 2021). Our setting differs in that we consider the meta-learning problem as an experimental design problem in which the relevant trade-off is between actively seeking information about the value of transferable or task-specific parameters. Although related, this is somewhat different from the exploration–exploitation dilemma considered by Simchowitz et al. (2021), where the learner trades off seeking information about the value

---

[3]Previous work has used the Laplace method (e.g., Ryan et al. (2015); Senarathne et al. (2020)) or stochastic gradient descent (Foster et al., 2019) to construct this approximation. Here, we use a moment-matching approximation in favor of the Laplace method since we observed that in our setting it leads to higher effective sample sizes.

[4]Although not shown explicitly in Equation (8) and Equation (9), each set of $M$ inner samples is constrained to include the corresponding sample $(\theta_i, \psi_i)$ to avoid pathological behavior when a value $y_i$ has low prior probability (Foster et al., 2020).

of parameters with reward maximization (see discussion in Section 3.2). Future work should better understand the connections between these two works and how the theoretical results of Simchowitz et al. (2021) can be leveraged in, e.g., hyperparameter selection for sOED-ML.

In the context of active learning, some work has provided theoretical results on the effect of (Cuong et al., 2016) and approaches to addressing (Go & Isaac, 2022) prior misspecification. Unlike this work, our focus here is on the application to the meta-learning context, i.e., to estimation of a subset of the parameters or of an embedded model (see discussion in Section 3.1). This connects our work to other problem formulations which constitute estimation of an embedded model. The remainder of this section discusses these connections.

**Model selection.** Applications of sOED to model selection, i.e., to identification of one of a set of models each characterized by a corresponding parameter distribution, is an example of an embedded model problem where the learner's goal is to identify the model indicator without particular regard for the corresponding parameter value (Foster, 2021). Sloman et al. (2023) showed that a phenomenon analogous to non-transferable information acquisition can occur in this context — misspecified parameter distributions can lead inference to favor the wrong model in early trials — and discussed the resulting model selection/parameter estimation dilemma, which can be seen as a special case of the meta-learning dilemma[5]. Past work has navigated the model selection/parameter estimation dilemma in the context of sOED by exclusive reliance on the EIG measure (i.e., computing the information gain measure with respect to both model and parameter values) (Borth, 1975) and alternating between maximizing information gain about the parameter values and model indicator (Cavagnaro et al., 2016).

**Optimization with Bayesian priors.** Bayesian optimization (BO) is an active learning method for maximizing black-box functions. BO can be seen as a case of sequential Bayesian active learning on an embedded model where the parameter of interest is the location of the function maximum (Hernández-Lobato et al., 2014; Foster, 2021). Prior work has shown that the choice of prior affects the performance of BO (Schulz et al., 2016). Solutions to this problem include online hyperparameter optimization (Berkenkamp et al., 2019), using prior data from experts to form a better prior (Wang et al., 2023), and constructing robust acquisition functions that explicitly account for the possibility of prior misspecification (Bogunovic et al., 2018; Kirschner et al., 2020).

---

[5]This connects closely with a body of work on the sensitivity of model selection indices to the choice of prior parameter distribution (Vanpaemel, 2010).

Multi-armed bandit problems are a related setting, which are instead characterized by a discrete action space. Theoretical results and bounds on the regret the learner should anticipate under prior misspecification have been established here (Kannan et al., 2018; Kveton et al., 2021; Bogunovic & Krause, 2021; Simchowitz et al., 2021). As mentioned above, many algorithmic solutions to prior misspecification in this setting promote "exploration" by incentivizing the learner to sample where they have high prior uncertainty (Kveton et al., 2021; Bogunovic & Krause, 2021; Simchowitz et al., 2021).

## 5. Experiments

To explore the behavior of sOED-ML, we ran a set of simulation experiments in an adapted version of the the preference modeling paradigm from Foster et al. (2019). Details of the paradigm can be found in Appendix D of their paper. Inputs were selected from among 20 scalar values of $x$ evenly spaced between $-80$ and $80$. These inputs are mapped to outputs $y$ such that the mean of $y \mid x$ depends on $x$ and $\theta$ (which is unknown to the learner), and the variance of $y \mid x$ depends on $|x|$ and $\sigma$ (which is known to the learner).

In the parameterization given in Foster et al. (2019), $\theta$ is the only parameter whose value is unknown to the learner. We consider this the transferable parameter, and include an additional, task-specific parameter $\psi$ that transformed inputs $x \rightarrow \psi x$. In other words, tasks characterized by high values of $\psi$ produced outputs with more extreme means and larger variances. Following Foster et al. (2019), we set the prior distribution of $\theta$ to $N(-20, 20)$ and the prior distribution of $\psi$ to $N(0, 1)$. In all experiments, $\theta^* = -12$ and $\psi^*$ was drawn from a true distribution $q(\psi)$. This distribution was $N(0, 1)$ in the well-specified case (i.e., matched the prior) and $N(2, 1)$ in the misspecified case. We ran both experiments where $\sigma = 1$ and where $\sigma = 5$.

Figure 3 shows the values of the acquisition functions discussed in Section 2 and Section 3 under the misspecified prior and $\sigma = 1$. In this example, choosing the value of $x$ that maximizes ETIG would results in non-transferable information acquisition, as indicated by the negative value of ATIG at the corresponding value of $x$.

When computing the values of our acquisition functions, we multiplied the covariance matrix of the best-fitting multivariate normal approximation to the posterior by a factor of 2. We set $N$ (the number of outer samples) to 10,000, and $M$ (the number of inner samples) to 100 (reflecting results from Rainforth et al. (2018) that $M$ is optimally $\propto \sqrt{N}$).

Figure 4 shows how the value of $\mathrm{CLB}_{\mathrm{ETIG}}$, computed according to different margins $\beta$ at the ETIG-maximizer, changes over the course of the experiments. When the margins are fixed (purple lines), the value of $\beta$ was set at

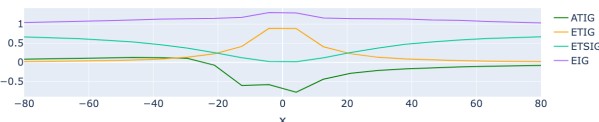

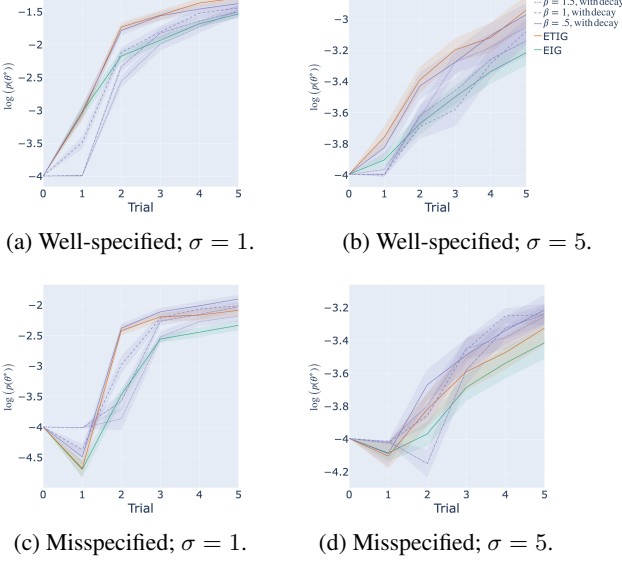

*Figure 3.* Values of the acquisition functions discussed in Section 2 and Section 3 in the context of the modeling paradigm used for our simulation experiments. Here, selecting the value of $x$ that maximizes ETIG would result in non-transferable information acquisition.

(a) Well-specified; $\sigma = 1$.  (b) Well-specified; $\sigma = 5$.

(c) Misspecified; $\sigma = 1$.  (d) Misspecified; $\sigma = 5$.

*Figure 5.* Degree to which each sOED method is able to recover $\theta^*$; sOED-ML is indicated by the corresponding decaying margin $\beta$. Lines indicate means and shaded regions indicate the corresponding standard errors across 100 simulated experiments.

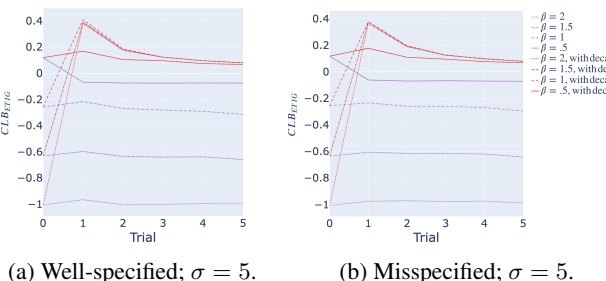

(a) Well-specified; $\sigma = 5$.  (b) Misspecified; $\sigma = 5$.

*Figure 4.* Value of $\mathrm{CLB}_{\mathrm{ETIG}}$ at the ETIG-maximizer on each trial of a sequential experiment, as computed according to the margins indicated in the legend. Lines indicate means and shaded regions indicated the corresponding standard errors across 100 simulated experiments.

the beginning of the experiment and remained unchanged. When the margins decay, the value of $\beta$ was set to the indicated value at the beginning of the experiment and decayed exponentially:

$$\beta_{t+1} = \frac{20\beta_t}{19 + 100\beta_t}. \tag{11}$$

Figure 4 shows that the fixed margins result in values of $\mathrm{CLB}_{\mathrm{ETIG}}$ that are relatively insensitive to where one is in the experiment: they remain persistently negative, indicating that they are dominated by inherent variance in the outputs and do not reflect the information the learner has gained about the value of $\psi$. The result is that sOED-ML never actively seeks transferable information. On the other hand, the decaying margins are negative at the beginning of the experiment, i.e., indicate the risk of non-transferable information acquisition that is reflected in Figure 3, but then increase and stabilize above 0, resulting in the acquisition of transferable information once the learner has gained information about the value of $\psi$.

Figure 5 shows how sOED-ML compares to sequential maximization of ETIG and EIG in terms of the learner's ability to recover $\theta^*$ (measured as $\log\left(p(\theta^*)\right)$; because of the pat-

terns described above, we show only results for sOED-ML using decaying margins). As expected, when the prior is well-specified (Figure 5a and Figure 5b), ETIG (orange line) outperforms all the other methods. However, this is not the case when the prior is misspecified (Figure 5c and Figure 5d): here, as anticipated, ETIG on average leads to non-transferable information gain (lower $\log\left(p(\theta^*)\right)$) after the first trial, which is protected against by some versions of sOED-ML.

## 6. Conclusion

In this paper, we leverage a Bayesian experimental design framework to analyze the setting of Bayesian meta-learning under prior misspecification. We expand upon existing measures of EIG and construct a distribution of information gain values, and then use this distribution to diagnose the risk of non-transferable information acquisition. As we demonstrated with our proposed algorithm sOED-ML, our work facilitates the development of active yet aware algorithms that perform well in varied task environments (such as when eliciting feedback from multiple experts). Future work can leverage the distribution of information gain values implicit in EIG measures in other ways, such as by developing acquisition functions for robust or "risk-seeking" experimental design.

## Acknowledgements

The authors thank Vikas Garg for helpful comments regarding the problem formulation, Avirup Das for helpful comments regarding the experimental implementation, and two anonymous reviewers for their feedback. SJS and SK were supported by the UKRI Turing AI World-Leading Researcher Fellowship, [EP/W002973/1]. AB was supported by the Academy of Finland Flagship programme: Finnish Center for Artificial Intelligence FCAI.

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

## A. Derivation of Equation (5)

$$
\begin{aligned}
\mathrm{AIG}(x) &= \mathbb{E}_{\omega \sim q(\omega)} \left[ \mathbb{E}_{y \sim q(y \mid x, \omega)} \left[ \log \frac{p(y \mid x, \omega)}{p(y \mid x)} \right] \right] \\
&= \mathbb{E}_{\omega \sim q(\omega)} \left[ \mathrm{H}\left( q(y \mid x, \omega) \mid\mid p(y \mid x) \right) - \mathrm{H}\left( q(y \mid x, \omega) \mid\mid p(y \mid x, \omega) \right) \right] \\
&= \mathbb{E}_{\omega \sim q(\omega)} \left[ \mathrm{H}\left( q(y \mid x, \omega) \right) + \mathrm{D}_{\mathrm{KL}}\left( q(y \mid x, \omega) \mid\mid p(y \mid x) \right) - \mathrm{H}\left( q(y \mid x, \omega) \right) - \mathrm{D}_{\mathrm{KL}}\left( q(y \mid x, \omega) \mid\mid p(y \mid x, \omega) \right) \right] \\
&= \mathbb{E}_{\omega \sim q(\omega)} \left[ \mathrm{D}_{\mathrm{KL}}\left( q(y \mid x, \omega) \mid\mid p(y \mid x) \right) - \mathrm{D}_{\mathrm{KL}}\left( q(y \mid x, \omega) \mid\mid p(y \mid x, \omega) \right) \right]
\end{aligned}
\tag{12}
$$

## B. Connections between $\mathrm{CLB}_{\mathrm{ETIG}}$ and Robust EIG

Go & Isaac (2022) developed a measure of *robust expected information gain (REIG)*, which can be interpreted as the lowest expected information gain from the set of distributions in an ambiguity set of radius $\epsilon$ around the learner's prior, i.e.,

$$
\mathrm{REIG}(x, \epsilon) = \inf_{Q} \left\{ \mathrm{EIG}(x, Q) \mid \mathscr{D}(Q, P) \le \epsilon \right\}.
\tag{13}
$$

Notice that the EIG is written as a function of both the input $x$ and the prior distribution $Q$ over which the infimum is taken; to compute the EIG as a function of a given $Q$, one replaces the probability measures indicated by $p$ in Equation (1) with those induced by $Q$. $P$ indicates some fixed reference prior, and $\mathscr{D}$ is a suitable divergence measure.

Notice that computing the REIG requires specifying the radius of the ambiguity set $\epsilon$. $\mathrm{CLB}_{\mathrm{ETIG}}$ admits a similar robustness interpretation as REIG, with the researcher degree of freedom being the choice of margin $\beta$ as opposed to the radius of the ambiguity set $\epsilon$. While Go & Isaac (2022) focus on a robust measure of the EIG, rather than the ETIG, they discuss extensions to their method for the case of a misspecified likelihood. Our discussion of the connection between REIG and $\mathrm{CLB}_{\mathrm{ETIG}}$ assumes this extended version of REIG as an information gain measure on the transferable parameter $\theta$.

Estimation of both REIG and $\mathrm{CLB}_{\mathrm{ETIG}}$ requires multiple samples of TIG from the learner's prior $p(\theta, \psi, y)$. In the context of REIG, changing the value of $\epsilon$ can be thought of as shifting the REIG between an average of these samples ($\epsilon = 0$; i.e., the ETIG) to the worst-case TIG from the sample ($\epsilon \to \infty$) (Go & Isaac, 2022). Similarly, in the context of $\mathrm{CLB}_{\mathrm{ETIG}}$, changing the value of $\beta$ can be thought of as moving the $\mathrm{CLB}_{\mathrm{ETIG}}$ from the sample average ($\beta = 0$; i.e., the ETIG) to a value of TIG $\beta$ standard deviations below this mean. In other words, higher $\beta$ moves the $\mathrm{CLB}_{\mathrm{ETIG}}$ further into the lefthand tail of an approximate distribution of TIG values. In the case where the true distribution of TIG values is continuous and the learner has access to an infinite number of samples from their prior, they can expect to recover values of $\mathrm{CLB}_{\mathrm{ETIG}}$ that are closer and closer to the worst-case TIG as $\beta \to \infty$. Thus, like increasing $\epsilon$ in the context of REIG, increasing $\beta$ shifts the $\mathrm{CLB}_{\mathrm{ETIG}}$ from the ETIG to the worst-case TIG value.