# OpenReview forum: "Bayesian Active Meta-Learning under Prior Misspecification"
_ICML.cc/2023/Workshop/ILHF — ILHF Workshop ICML 2023_

### Official Review · Reviewer_c8Ry · 2023-06-09
**Review for Submission 16**

**Rating:** 6
**Confidence:** 3

**Review:**

## Summary

This paper introduced and studied the dilemma in meta learning about whether the agent should extract transferable information
based on their prior beliefs, which is possibly misspecified, or seek task-specific information that enables better identification of the current task environment. The authors formulated the problem from mutual information perspective and proposed methods to diagnose the risk of non-transferable information acquisition and use it to nevigate the meta-learning. They also provided empirical verification for their methods.

## Comments
The paper is relatively clear to follow.

I'm not sure if the "dilemma" is the best way to formulate the problem. I think the agent should always first try to identify the current task environment first (i.e. fix the prior in example of Fig 2) before solving \theta, and the only thing matters is how much effort is required for that fixing process.

There are some approximation steps in the derivation of CLB, which itself is also very heuristic. Although some empirical verification is provided, it's unclear when it is a good approximation and when it will fail. (But I understand that some difficulty is fundamental)

---

### Official Review · Reviewer_8vzi · 2023-06-14
**Interesting Topic, Limited Experimental Evaluation**

**Rating:** 7
**Confidence:** 3

**Review:**

**Paper Summary**
This paper examines the problem of Bayesian Active Meta-Learning. They consider that the parameter of interest can be partitioned into a transferable component of interest and a task-specific component. Intuitively, in the misspecified prior setting, the algorithm’s expectation of the information gain over the transferable component can diverge from the true gain (thereby causing some non-transferable information to be incorporated into the transferable parameter). Countering this can require investing in the disambiguation of the current non-transferable parameter (which affects the information gain calculation) but this must be appropriately balanced with learning the actual parameter of interest. This paper provides a credible lower bound over the actual information gain w/r/t the parameter of interest and uses it to trade-off optimizing the non-transferrable reward (avoiding the divergence between the true and actual gain of the parameter of interest) and actually disambiguating the parameter of interest.

**Strengths**
This paper introduces and carefully analyzes a dilemma that can arise in Bayesian Active Meta Learning when there is prior misspecification regarding the extraction of task-independent information. The setting proposed here does seem to be distinct from prior work, though this can be clarified a little better in the text. They present this dilemma clearly, using both numerical examples and theory, and the quality and quality of the writing is very good. Their proposed method is well-motivated by the formulation of the problem and they draw conceptual connections to exploration and robustness-style approaches (both of which are interesting).

Weaknesses
The experiments are conducted in only a single toy domain which does present important limitations for establishing the utility of this method, as well as the extent to which the dilemma arises under real-world conditions. The nexus to this workshop’s topic is presented as learning from a population of human users (with varying idiosyncrasies/biases), which is plausible, but it would have been nice to have a small user study to illustrate this. I believe that a user study would be especially helpful given that some approximations are needed to ensure the tractability of the method and these might need to be verified on real-world data.

Even if a user study is not possible, it would be nice to show a concrete and real-world example of Bayesian meta-learning in which non-transferable information results in poor estimates of the transfer of parameters. This could still be done in simulation but in a more realistic fashion than the current experiments. For example, maybe involve known models of human biases from the cognitive science literature.

**Questions/Smaller Concerns**
1.The discussion of Figure 2 in Section 1.2 is a little confusing at first glance. For example, the notation y|x=1, corresponds to observing that y=1  under the distribution y|x (correct?) but it looks like it could be y|(x=1). Similarly, I found the shading to be confusing in Figure 2.

2.In related works, there is some discussion of the exploration/exploitation problem in contextual bandits and the text mentions, “Although related, this is somewhat different from the exploration–exploitation dilemma”. It would be nice to have some more explicit statements on both sides. My guess is that the “relation” arises from the exploration analogy to the method introduced in this work and the “difference” arises from parameter estimation vs. reward maximization.

3.In the experiments section graphs (Fig 5), some recoloring of the lines might be nice to make it easier to read.

**Conclusion**
Ultimately, despite my desire for some more in-depth experimental evaluation of the proposed method and setting, I believe that this is a well-written paper that addresses a novel and interesting problem. As such, I am in favor of acceptance.

---

### Decision · Program_Chairs · 2023-06-20

Accept